# Learning to Correct Mistakes: Backjumping in Long-Horizon Task and Motion Planning

**Yoonchang Sung**[1*], **Zizhao Wang**[1*], **Peter Stone**[1,2]
[1]The University of Texas at Austin [2]Sony AI

**Abstract:** As robots become increasingly capable of manipulation and long-term autonomy, long-horizon task and motion planning problems are becoming increasingly important. A key challenge in such problems is that early actions in the plan may make future actions infeasible. When reaching a dead-end in the search, most existing planners use backtracking, which exhaustively reevaluates motion-level actions, often resulting in inefficient planning, especially when the search depth is large. In this paper, we propose to learn backjumping heuristics which identify the culprit action directly using supervised learning models to guide the task-level search. Based on evaluations on two different tasks, we find that our method significantly improves planning efficiency compared to backtracking and also generalizes to problems with novel numbers of objects.

**Keywords:** Task and motion planning, heuristic learning, supervised leanring

## 1 Introduction

Integrated task and motion planning (TAMP [1]) is a framework for making sequential decisions in robotic tasks. Solving TAMP problems involves a hybrid search over a sequence of discrete actions (*e.g.*, which object to manipulate) and their continuous motion parameters (*e.g.*, with which pose to grasp the object). One important practical challenge is that for long-horizon tasks with a large number of objects, the search space becomes intractable due to a large depth and branching factor.

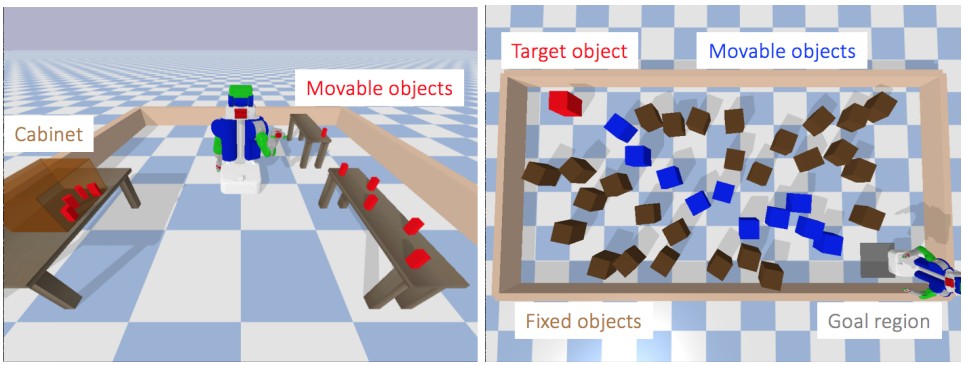

Figure 1: (Left) An illustration of the packing task, whose goal is to move all red objects into the cabinet. Putting first several objects near the entrance prevents the robot from putting the remaining objects inside. (Right) An illustration of the navigation among movable obstacles (NAMO) task whose goal is to move the red target object to the goal region.

In this work, we address the challenges of long-horizon TAMP where early actions in the plan may make future actions infeasible. For example, as shown in Fig. 1, if the robot places the first few objects near the cabinet entrance in a packing task, or on the target retrieval path in a navigation among movable obstacles (NAMO [2]) task, the problems become infeasible, unless the culprit actions that

---

[*]Equal contribution.

6th Conference on Robot Learning (CoRL 2022), Auckland, New Zealand.

placed objects in obstructive positions are corrected. When planning fails, identifying culprit actions is desirable as the search can focus on correcting the culprit actions, ignoring irrelevant actions.

Generating explanations for failure has been explored to guide search [3, 4, 5]. In TAMP, geometric failures guide the task-level planner, often done via backtracking to try a different alternative for the immediately prior action in the plan [6]. As backtracking exhaustively explores the search tree (Fig. 2(b)), correcting the culprit action can involve an exponential number of evaluations of all the intermediate actions. The complexity becomes especially daunting in TAMP because the continuous motion parameters induce an infinite branching factor for each action.

In the constraint satisfaction literature [7], *backjumping* has been introduced to alleviate the complexity of backtracking by taking a short cut to an ancestor action. In discrete settings, it may be possible to jump back to the culprit action directly without precluding any possible solutions. However, because of the infinite branching factor in TAMP, conclusively identifying the culprit action is generally not possible. Even estimating the culprit requires solving the original backtracking problem, which can be very time-consuming (see details in Sec. 4).

We thus propose a learning approach by shifting the computational burden to the training phase where training data is collected by solving backtracking problems. Our observation is that because the true culprit action always exists in any failure cases, learning models designed to exploit this may predict the culprit accurately. Specifically, we explore two frameworks to learn backjumping heuristics: (1) *imitation learning* which directly predicts the culprit from all previous actions, and (2) *plan feasibility* which indirectly predicts the culprit by checking if the solution can be found after each action. Our proposed learning process is a domain-independent way of learning domain-specific heuristics. We summarize our contributions as follows:

- We propose two backjumping heuristic learning methods to improve the efficiency of solving long-horizon TAMP problems. We also present two algorithms containing learning methods as a subcomponent, which treats the continuous parameters differently.

- Our empirical results show that our methods improve planning efficiency by 40% for packing and 99% for NAMO, both against backtracking. Furthermore, by incorporating a graph neural network (GNN [8]) into the learning model, our methods can generalize to problems with novel numbers of objects while outperforming backtracking consistently.

## 2 Preliminaries

We introduce the TAMP notation that will be used to define our problem and the sequence-before-specify strategy that we improve on.

### 2.1 G-TAMP notation

Although our method can potentially be applied to a broader class of TAMP problems [1], we focus on a particular subclass of TAMP called *geometric TAMP* (G-TAMP [9]) for clarity of presentation. In G-TAMP, a mobile manipulator is tasked with moving multiple objects to target regions among movable obstacles. We assume quasi-static dynamics of the world[1], deterministic effects of actions, and fully-observable environments. Even with these assumptions, the proposed problem is already hard (NP-hard explained in Sec. 7), but relaxing these assumptions may still be possible by introducing belief-space planning [10, 11], which we leave as future work.

Formally, a G-TAMP problem is defined by a tuple $< \mathcal{M}, \mathcal{R}, \mathcal{O}, F, T, \mathcal{G}, s_0 >$ where $\mathcal{M}$ is a set of *movable objects*, $\mathcal{R}$ is a set of *target regions*, $\mathcal{O}$ is a set of *operators*, $F$ is a *feasibility checking function*, $T$ is a *transition function*, $\mathcal{G}$ is a *goal set*, and $s_0$ is an *initial state*. The goal is to find a sequence of grounded and refined operators[2] starting from $s_0$ to reach a state $s \in \mathcal{G}$.

• $\mathcal{M}$, $\mathcal{R}$ and $s$: In G-TAMP, the environment consists of any number of *fixed objects* such as tables, $N_\mathcal{M}$ movable objects $\mathcal{M} = \{m_i\}_{i=1}^{N_\mathcal{M}}$ such as cups, and $N_\mathcal{R}$ target regions $\mathcal{R} = \{r_j\}_{j=1}^{N_\mathcal{R}}$. We denote the *world state* by $s$ which consists of the robot configuration and the pose of each movable object.

---

[1]The objects remain in stable states after being manipulated by the robot.
[2]We say that an operator is *grounded* if its discrete parameters are specified and *refined* if its continuous parameters are specified.

- $\mathcal{O}$: Operators in G-TAMP move one object to a region, such as PICKANDPLACE and PUSH. Each refined operator $o\,(d=(m_i,r_j),c=(q,g))\in\mathcal{O}$ consists of (1) the *discrete parameter* $d\in\mathcal{M}\times\mathcal{R}$ specifying which object to move and which region to move it to, and (2) the *continuous parameters* $c$ composed of a grasp $g\in SE(3)$[3] on $m_i$ and a placement pose $q\in SE(3)$ in $r_j$.

- $F$ and $T$: The feasibility of an operator can be determined by evaluating constraints (*e.g.*, reachability and collision avoidance) using an external motion planner. We denote this feasibility evaluation as a Boolean-valued function $F(s,o)\to\{0,1\}$. If the operator is feasible, a deterministic transition function $T(s,o)$ determines the new state $s'$.

- $\mathcal{G}$: A goal set $\mathcal{G}$ is defined as a conjunction of predicates, such as INCABINET $(m_i,r_j=$ CABINET$)$ which becomes TRUE if $m_i$ is stably located in the cabinet.

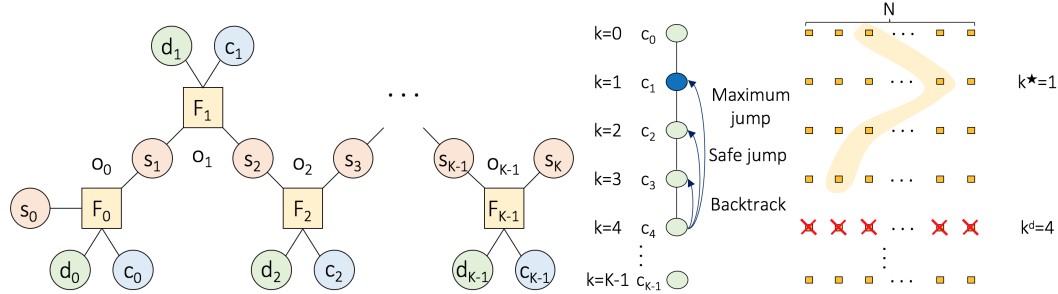

Figure 2: (a) Constraint network consisting of variables represented by circles (continuous variables: $s$ and $c$, and discrete variables: $d$) connected by constraints $F$ represented by squares. (b) Example of a search tree when a dead-end is met at level 4 (*i.e.*, $k^d$). Orange squares represent sampled values ($N$ values at each level). Orange shaded area denotes a particular assignment to $(c_0,...,c_3)$ (*i.e.*, a partial plan). All values of $c_4$ are inconsistent with this partial plan causing the dead-end. Maximum jump $k^\star$ in this example is 1 (a culprit variable is $c_1$ denoted by a blue circle), and thus, $\mathcal{K}=\{1,2,3\}$.

G-TAMP problems can alternatively be seen as hybrid constraint satisfaction problems, as shown in Fig. 2(a). Circles represent *variables* whose domains are either continuous ($s_k$ and $c_k$) or discrete ($d_k$), and squares represent *constraints* ($F_k$). When the plan length is $K$, we find values for sets of variables $\{d_k\}_{k=0}^{K-1}$ and $\{c_k\}_{k=0}^{K-1}$ such that all constraints $\{F_k\}_{k=0}^{K-1}$ are satisfied and $s_K\in\mathcal{G}$.

## 2.2 Sequence-before-specify strategy

Sequence-before-satisfy [4, 12, 13, 14, 15, 16, 17, 1] is one strategy to solve TAMP problems which this work is based on. The strategy consists of two stages: (1) In the sequencing stage, the strategy finds *plan skeleton*s [18] reaching the goal set $\mathcal{G}$ symbolically only, *i.e.*, $(d_k)_{k=0}^{K-1}$ are computed but their continuous counterparts $(c_k)_{k=0}^{K-1}$ are left unspecified. (2) In the specifying stage, the strategy chooses a plan skeleton often based on AI planning heuristics and *refine*s $(c_k)_{k=0}^{K-1}$ satisfying the constraints and a goal condition. For notational simplicity, in the rest of the paper, we use $m_k$ to represent the object moved by operator $o_k$. We treat objects that are not included in a grounded plan as fixed objects.

Note that the two stages repeat until either finding a solution or reporting no solution, generating multiple plan skeletons. Our method (Sec. 4 and Sec. 5) accommodates these multiple plan skeletons by learning a specific model for each plan skeleton. In this section we focus on the case of a single plan skeleton for ease of presentation.

Backtracking search is generally used to find values of $(c_k)_{k=0}^{K-1}$ in the specifying stage by constructing a search tree rooted from $c_0$ (Fig. 2(b)). Since the domain of $c_k$ is continuous, the strategy uses sampling (*e.g.*, uniform sampling from the domain of $c_k$ [19]) to obtain $N$ refined values to select from. Index $k$ corresponds to both the level of the tree and the step in the plan skeleton. When the assigned values of $(c_0,...,c_{k-1})$ are inconsistent with all sampled values of $c_k$ (*i.e.*, violating the

---

[3]The special Euclidean group $SE(3)$ is used to express a 3D rigid-body transformation consisting of translation and rotation.

constraint $F_k$), we say the search hits a *dead-end* and denote the dead-end level by $k^d$. The search then backtracks to a parent node $c_{k-1}$ to assign another value and retries the consistency check with $c_k$. If successful, a child node $c_{k+1}$ is evaluated at the next level, otherwise another value of $c_{k-1}$ is attempted. Backtracking repeats this process exhaustively until it finds values of $(c_k)_{k=0}^{K-1}$ that are consistent, *i.e.*, the solution of a problem.

## 3 Problem Description

When the planning horizon is long, the above backtracking-based strategy becomes intractable as the search space increases exponentially. Our goal in this work is to adopt the idea of backjumping [7] to effectively reduce the search space and achieve efficient planning in TAMP.

In contrast to backtracking which only backtracks level by level when dead-ends persist, we can speed up the search by backjumping multiple levels (Fig. 2(b)). A backjump is said to be *safe* if it does not preclude any solutions. Specifically, at level $k^d$, *safe jump* corresponds to an ancestor level $k < k^d$ such that attempting all possible values of its descendants at all levels between $k$ and $k^d$ does not resolve the dead-end. We denote all safe-jump levels by a set $\mathcal{K}$.

The larger the jump, the better, because larger jumps avoid more computation. We denote the largest safe jump by *maximum jump*, *i.e.*, $k^\star = \min \mathcal{K}$. Correspondingly, we call $c_{k^\star}$ a *culprit variable* responsible for making all values of $c_k$ being inconsistent with a partial plan.

The objective is to find $k^\star$ to backjump safely and maximally whenever a dead-end is met at level $k^d$, thus improving the overall efficiency of the sequence-before-specify strategy.

## 4 Backjump Learning Methods

The constraint network in Fig. 2(a) contains a chain structure with respect to constraints $F_k$ by considering $d_k$, $c_k$, and $s_k$ as one large variable node for each $k$. Because of this topological structure, no constraints connect variables that are more than one level apart. Moreover, we can only estimate $k^\star$ as we use sampling to handle continuous domains. For these reasons, $k^\star$ identification using backjumping is inherently challenging in TAMP.

It is however possible to find an approximation of $k^\star$ with backtracking since it evaluates all possible combinations of assignments using all sampled values from the root to level $k^d$. Also, since $k^\star$ always *exists* for any dead-end situations and is *unique*, we can hope to gather less noisy labeled data with a sufficient amount of samples. Leveraging these observations, we propose two alternative supervised learning methods to identify $k^\star$, where training labels are gathered by solving tasks from the same distribution using backtracking.

The robot configuration is used in an external motion planner to check the feasibility of a corresponding refined operator, but it is not used in our learning models. We thus define a new state notation $\bar{s}$ consisting of poses of movable objects only.

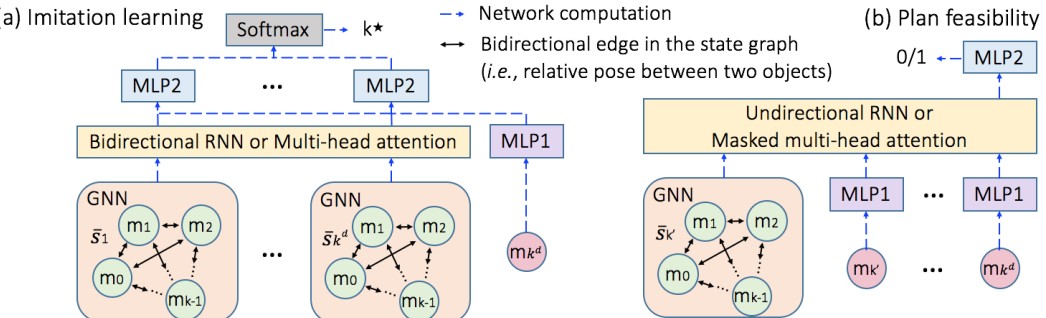

Figure 3: Architectures of the learning models. In each graph, state $\bar{s}_k$ implies that objects $(m_0, ..., m_{k-1})$ are relocated by the robot while other objects $(m_k, ..., m_K)$ remain in their original locations.

## 4.1 Imitation learning

When the search hits a dead-end at level $k^d$, we first propose a predictor to directly predict the maximum jump $k^\star$ in the domain of $\{0, \ldots, k^d - 1\}$, from the sequence of states $(\bar{s}_1, \ldots, \bar{s}_{k^d})$ and the geometric attributes of a movable object $m_{k^d-1}$ (*e.g.*, size). In our model, each state $\bar{s}$ is represented by a fully-connected graph whose nodes are the poses of movable objects and each edge is the relative pose between each pair of objects.

As shown in Figure 3(a), to predict $k^\star$, our model (1) extracts the state features from each state in the sequence using the same GNN, (2) computes the temporal features across the sequence of state features using a bidirectional recurrent neural network (bRNN [20]) or multi-head attentions [21], (3) extracts the movable object features from $m_{k^d}$ with MLP1 (multilayer perceptron), and (4) with pairs of temporal features and object features as inputs, applies the same prediction network MLP2 to predict the likelihood of each step being $k^\star$. The model is trained to minimize the cross-entropy loss between the predicted likelihood of being $k^\star$ and the ground-truth $k^\star$ label.

The $k^\star$ labels are collected from the backtracking tree search. At any $k^d$, we record its current state trajectory as $(\bar{s}_1, \ldots, \bar{s}_{k^d})$. When the backtracking successfully reaches the level $k^d$ for the first time after the dead-end, we record its state trajectory up to level $k^d$ as $(\bar{s}'_1, \ldots, \bar{s}'_{k^d})$. Let $\bar{s}_i$ and $\bar{s}'_i$ be the $i$-th element from the corresponding trajectory sequences. The $k^\star$ is the first level where two trajectories diverge, *i.e.*, $k^\star = \arg\min_i \{i \in \{0, ..., k^d - 1\} \big| \bar{s}_{i+1} \neq \bar{s}'_{i+1}\}$.

## 4.2 Plan feasibility

Besides directly estimating $k^\star$, we investigate an alternative *counterfactual* approach that learns a binary classifier to predict whether a refined partial plan at each level contributes to a dead-end. We use the predicted labels (*i.e.*, either feasible or infeasible) to identify which level corresponds to $k^\star$.

Specifically, with a sequence of states $(s_1, \ldots, s_{k^d})$ that faces a dead-end at level $k^d$, we start with $s_1$ where only $c_0$ is refined (*i.e.*, a value is assigned) but the rest of the variables $(c_k)_{k=1}^{k^d}$ are not refined yet.[4] Then, the classifier predicts whether finding a consistent assignment of values for $(c_k)_{k=1}^{k^d}$ is feasible. If infeasible, it implies that the placement of $m_0$ is a culprit action making the rest of variables $(c_k)_{k=1}^{k^d}$ inconsistent with $c_0$. By definition, level 0 becomes the maximum jump $k^\star$. If feasible, we continue with $s_2$ where $c_0$ and $c_1$ are refined only and the classifier predicts feasibility for $(c_k)_{k=2}^{k^d}$. Likewise, we iteratively apply the classifier to predict feasibility in an ascending order of $k$; we stop if infeasibility is predicted and output the corresponding step as $k^\star$, or continue to the next step otherwise.

Let $k' \in \{1, ..., k^d\}$ be the step where $(c_k)_{k=0}^{k'-1}$ are refined and feasibility for $(c_k)_{k=k'}^{k^d}$ is to be evaluated. As shown in Figure 3(b), to predict whether a level $k' < k^d$ is a safe jump, our model (1) extracts the state features from $\bar{s}_{k'}$ using GNN, (2) extracts the object features from $(m_k)_{k=k'}^{k^d}$ each with MLP1, (3) computes the temporal feature from the state features and object features using a unidirectional RNN or multi-head attention (the inputs to the attention are also masked in a way to make the computation unidirectional) and (4) applies the classification network MLP2 to the temporal feature to predict whether finding consistent values for $c_{k^d}$ is feasible.

For model training, note that plan feasibility can be trained on any pair of state $s_{k'}$ and future variables $(c_{k'}, \ldots, c_k)$ where $c_k$ does not have to be a dead-end, thus making it relatively easy to generate a large amount of training data. For any state $s_{k'}$ (*i.e.*, assigned values of $(c_k)_{k=0}^{k'-1}$ in the search tree), if its subtree reaches a level $k \geq k'$, the plan feasibility for $(c_{k'}, \ldots, c_k)$ is 1, or 0 otherwise. The model is trained to minimize the binary cross-entropy loss between the predicted and ground-truth feasibility for $(c_{k'}, \ldots, c_k)$.

## 5 Backjumping Algorithms

In this section, we present our algorithm that leverages the proposed learning methods as backjumping heuristics to guide the search. The overall algorithm is similar to backtracking except that, at dead-ends, backtracking is replaced by backjumping with $k^\star$ predicted by the trained model.

---

[4] Note that refining $c_k$ determines the pose of a movable object $m_k$.

Since we sample a finite number of values for $(c_k)_{k=0}^{K-1}$ from their continuous domains, we need a mechanism to sample more values if a solution is not found with those currently available values. We propose two versions of the algorithm: (1) the *batch sampling* method, and (2) the *forgetting* method, differing by how the sampling process is treated. The pseudocodes of both algorithms are presented in the appendix.

In batch sampling, we first draw a batch of $N$ samples for each variable of $(c_k)_{k=0}^{K-1}$, and then find a consistent sequence of values using the backjumping algorithm. If a solution is not found before the search tree is exhausted, we draw another batch of samples to construct a new search tree. Since the search tree maintains the same set of values, memoization techniques such as *no-goods*[5] can be applied to further accelerate the search, which we leave as future work.

The forgetting method on the other hand does not keep previously sampled values, but discards them and redraws new samples each time the search visits a different level in the search tree. Thus, memoization is unavailable in the forgetting method, although it may potentially explore the continuous search space effectively.

Both algorithms are run until the time limit is reached. It is an open question which of the two methods is more theoretically beneficial (*e.g.*, in terms of convergence rate); we instead show the performance of both methods empirically.

## 6 Evaluation

This section reports on experiments designed to evaluate the following hypotheses: (1) How *efficiently* can our backjumping methods find a solution in comparison with backtracking? (2) How well does our model *generalize* to different numbers of objects? (3) Which of the backjumping algorithms between batch sampling and forgetting performs more efficiently? We evaluate these hypotheses on the packing and NAMO tasks, as described next.

### 6.1 Evaluation tasks

Our evaluation tasks are implemented in the PyBullet [22] simulator where a PR2 robot is used as a mobile manipulator (see Fig. 1). We use PDDL [23, 24] to find plan skeletons in the sequencing stage and bidirectional RRT [25] as an external motion planner [26] in the specifying stage.

In the packing task, the objective is to move all objects located randomly on the right tables into the cabinet on the left. For a PICKANDPLACE operator where the target region $r_j$ is the cabinet, we define the sampling domain as the 2D base plane of the cabinet to draw placement pose samples $p$. As the interior space of the cabinet can be accessed from one side only and is not spacious enough, objects placed near the entrance may make placing future objects infeasible.

In the NAMO task, there are 10 movable boxes (in blue) located randomly in the vicinity of the robot's path and 27 fixed boxes (in brown) in the room and the goal is to move the target box (in red) to the goal region (in gray). As blue boxes block the path to reach the target box, the robot must clear them by relocating them in their vicinity. We also define the domain for sampling box placement poses $p$ from a circular arc on the floor computed with respect to the robot base frame. When moving back to the goal region, with the target box in hand, the robot is unlikely to find a feasible motion plan if blue boxes were relocated to blocking positions. Unlike the packing task where objects move from one fixed object to another fixed object, objects are moved to and from the same fixed object (*i.e.*, the floor) in NAMO.

### 6.2 Implementation details

For both backtracking and backjumping, we use $N = 30$ for the packing task and $N = 4$ for the NAMO task. We denote our method and architecture combinations as:
• **IL-RNN** and **IL-Attn**: Imitation learning with RNN and with multi-head attention, respectively.
• **PF-RNN** and **PF-Attn**: Plan feasibility with RNN and with multi-head attention, respectively.
The architecture and training details can be found in the appendix. For PF, as the trained model outputs a probability of being either feasible or infeasible, we additionally introduce a threshold $\epsilon$

---

[5]No-good is an assignment to a subset of $\{c_k\}_{k=0}^{K-1}$ that cannot be extended to any solutions.

to make a classification decision robust. We apply the model to each step and record a predicted probability of being feasible at step $k$ as $p_k$. We select $k^\star$ as the first step whose $p_k < \epsilon$. $\epsilon$ is computed by averaging between the maximum and minimum values of $p_k$, where $k$ is an element from $(0, \ldots, k^d - 1)$. Empirically, we find that introducing this adaptive threshold $\epsilon$ predicts $k^\star$ more accurately than a fixed threshold, *e.g.*, setting $\epsilon = 0.5$.

All our reported results are obtained by training with 3 different seeds. We observe in our supervised learning setting that changing the seed to train the model does not affect the performance much.

In Sec. 6.5, we show that the forgetting algorithm empirically outperforms batch sampling. Thus, we use the forgetting algorithm to obtain results for planning efficiency and generalization. We include complete results for batch sampling in the appendix.

## 6.3 Planning efficiency

We collect data from 500 problems with 10 objects for packing and from 250 problems for NAMO. New 100 problems and 50 problems are used to test packing and NAMO, respectively.

To measure planning efficiency, we consider the number of nodes visited in the search tree, where fewer nodes imply fewer feasibility checks, leading to faster planning. As shown in the top rows of Table 1, in both tasks, our backjumping methods are significantly more efficient than backtracking. We also show how closely our backjumping $\hat{k}^\star$ approximates the ground-truth $k^\star$ in the appendix.

In NAMO, imitation learning outperforms plan feasibility. We conjecture that movable objects in NAMO do not affect each other (*i.e.*, relocating one box does not affect the feasibility of relocating another box), making it easier for imitation learning to find $\hat{k}^\star$ close to $k^\star$.

We also show in the appendix that some methods still greatly outperform backtracking with fewer training data, and that the performances in Table 1 can further be improved with more training data.

Table 1: The number of nodes visited in the search tree. Numbers represent the mean $\pm$ 95% confidence interval computed by solving 100 problems. In the middle rows, numbers in $(\cdot)$ represent the number of objects tested. In the bottom row, **BS** represents batch sampling. The number with $*$ represents the best-performing method. The bolded numbers are those whose performance is not statistically significantly different from the one with $*$ (*i.e.*, their confidence intervals are overlapping).

| Task | Backtracking | IL RNN | IL Attn | PF RNN | PF Attn |
|------|--------------|--------|---------|--------|---------|
| Packing | $4414 \pm 879$ | $\mathbf{2464 \pm 464}$ | $\mathbf{2638 \pm 602}$ | $\mathbf{2205 \pm 313}$ | $\mathbf{2062 \pm 297}^*$ |
| NAMO | $(21 \pm 10) \times 10^4$ | $\mathbf{543 \pm 187}$ | $\mathbf{425 \pm 153}^*$ | $\mathbf{529 \pm 188}$ | $2614.7 \pm 709.5$ |
| Packing (11) | $12098 \pm 2518$ | $\mathbf{5350 \pm 1094}^*$ | $\mathbf{7044 \pm 1481}$ | $\mathbf{6142 \pm 767}$ | $7109 \pm 809$ |
| Packing (12) | $34719 \pm 6514$ | $\mathbf{15139 \pm 3080}^*$ | $\mathbf{16339 \pm 3971}$ | $22377 \pm 3244$ | $31824 \pm 3925$ |
| Packing (**BS**) | $13541 \pm 4205$ | $\mathbf{4464 \pm 1160}$ | $7073 \pm 2040$ | $\mathbf{4556 \pm 749}$ | $\mathbf{4311 \pm 690}^*$ |

## 6.4 Generalization

We examine whether our method can generalize to a novel number of objects. In particular, we train our model using 10 objects and test with more objects (*i.e.*, 11 and 12 objects). Note that adding more objects to the packing task is more challenging than removing objects as it leads to more dead-ends and harder estimation of $k^\star$, due to insufficient space in the cabinet.

As shown in the middle rows of Table 1, our methods still significantly outperform backtracking and the performance ratios of our methods and backtracking remain similar to those in the top rows where the number of objects is the same for both training and testing, with the only exception of **PF Attn** in Packing (11) and (12). We conjecture that the use of GNNs allows our learning models to handle novel numbers of objects.

## 6.5 Comparison of backjumping algorithms

As shown in the bottom row of Table 1, evaluated on the same set of 100 problems in packing with 10 objects, batch sampling is about $2 \sim 3$ times slower than forgetting, while still outperforming

backtracking. Although forgetting empirically exceeds batch sampling in our evaluation, we point out in Sec. 5 that enhancing batch sampling is potentially achievable by leveraging ideas from the constraint satisfaction literature [7].

# 7   Related Work

In TAMP [27, 10, 28, 4, 16], geometric failures are often used to provide feedback to task-level planning to guide its search. However, this feedback in literature is mostly *local* [5] meaning that the failure is involved with a particular sequence of actions. In this work, we propose a *global* feedback that considers the entire plan skeleton by leveraging backjumping.

The minimum constraint removal (MCR) problem [3] is related to our problem as its objective is to find a parsimonious set of objects to remove in order to reach a goal region. Our problem is a generalization to MCR in that a goal placement of an object is not defined and that objects are not removed but relocated. Since MCR is NP-hard, our problem also falls into the NP-hard category.

The most relevant papers to our work are the culprit detection problem [5] and the work [12]; both papers address shortcomings of backtracking. A method proposed in the culprit detection problem generates global feedback from failures in answer set programming [29] but it pre-discretizes the state which may ignore certain motion constraints. The work [12] improves the efficiency of back-tracking by jointly optimizing parameters such as trajectory, grasp, and placement pose; their work uses trajectory optimization, which is a competing framework for sampling-based motion planning our method is based on.

Learning other heuristics to improve planning efficiency has been proposed in the TAMP and motion planning communities, such as predicting feasibility at the motion level [30, 31, 32] and guiding sampling [33, 34]. Instead, our approach learns task-level heuristics by analyzing *long-horizon dependencies*, which can potentially complement existing heuristics to improve efficiency further.

# 8   Limitations

Besides future work discussed in the paper, some limitations of the presented work are as follows:
(1) We consider incorporating backjumping with a given single plan skeleton only, not with a set of possible plan skeletons, which may be critical when objects have substantially different sizes.
(2) We test on relatively simple geometric objects and assume known 3D CAD models. A more realistic geometry of objects, *e.g.*, constructed by point clouds, in our framework, is a future direction.
(3) We evaluate with the same geometry of obstacles (*i.e.*, the same cabinet in packing and the same arrangement of fixed objects in NAMO). Future work would be to achieve generalization to novel geometry of obstacles drawn from some distribution.

# 9   Conclusion

In this paper, we present learning frameworks to learn backjumping heuristics from data to identify the culprit action aimed at improving the efficiency of solving TAMP problems. Our experimental results demonstrate our method exceeds backtracking by far in terms of planning efficiency and its generalization to problems with novel numbers of objects.

**Acknowledgments**

This work has taken place in the Learning Agents Research Group (LARG) at UT Austin. LARG research is supported in part by NSF (CPS-1739964, IIS-1724157, FAIN-2019844), ONR (N00014-18-2243), ARO (W911NF-19-2-0333), DARPA, GM, Bosch, and UT Austin's Good Systems grand challenge. Peter Stone serves as the Executive Director of Sony AI America and receives financial compensation for this work. The terms of this arrangement have been reviewed and approved by the University of Texas at Austin in accordance with its policy on objectivity in research.

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
