# OpenReview forum: "Learning to Correct Mistakes: Backjumping in Long-Horizon Task and Motion Planning"
_robot-learning.org/CoRL/2022/Conference — CoRL 2022 Poster_

### Official Review · Reviewer_q4VZ · 2022-07-20

**Originality:** Good
**Technical Quality:** Good
**Clarity Of Presentation:** Good
**Impact:** 3

**Recommendation:**

Weak Accept: I recommend accepting the paper, but will not argue for my recommendation if the majority of other reviewers have a different opinion.

**Summary:**

This paper considers the backtracking search over action refinements that must be performed in a sequence-then-refine Task and Motion Planning solver, after a promising action skeleton has been identified.

If a mistake is made is made early on in refinement, the search will eventually reach a dead-end. If the number of steps between the "culprit" action and the dead-end is large, then backtracking will waste lots of computation on sampling different refinements of the intermediate steps.

This paper proposes to address this problem by learning a GNN-based model to predict the culprit after encountering a dead-end. This model can then be used to "backjump" directly to the culprit, without sampling refinements for intermediate actions. The model is trained using supervised learning with labels obtained from the backtracking algorithm.

The authors evaluate the approach in two simulated robotics problems, and demonstrate improvements in search efficiency over backtracking.

**Issues:**

* The evaluation tasks are not well described. How exactly do they differ from each other, and from the training data. In the packing task, are all object locations sampled at random. In the NAMO are all the movable objects' locations randomly sampled? These details should be spelled out.

* (Clarified in rebuttal) It would be interesting to get a sense of the overhead of querying these models. Since PF models are being queried at every step, I imagine this would be quite considerable. In comparison, the IL models seem to only need to be queried once per dead-end. But beyond comparing the different architectures, it would be crucial to compare the time we save on backtracking with model inference time.




**Quality Of The Limitations Section:**

Additional details required

**Reviewer Expertise:**

4: The reviewer is confident but not absolutely certain that the evaluation is correct

**Robotics Focus:**

Highly relevant to robotics but no hardware experiments

**Strengths And Weaknesses:**

Strengths

* Culprit identification is difficult to get at analytically, so a learning method seems promising, and potentially very useful. The paper also makes a nice connection to the constraint satisfaction literature’s concept of backjumping.

* The learning approach and model architectures seem reasonable, and the results on evaluation tasks demonstrate a significant improvement over backtracking.

Weaknesses:

* (Improved in rebuttal + to be added to paper) In general, I think the empirical evaluation of the method is lacking. At the moment, it is hard to glean any insight beyond the fact that this approach exhibits improved sample efficiency than vanilla backtracking on two problem types. Some other baselines/ablations would help - for instance what if backtracking just jumped a constant 2-3 steps back every dead-end. How would that compare?
* (Improved in rebuttal + to be added to supplementary materials) There is no discussion or analysis of the overhead that model inference adds to the backtracking procedure.
* It is not clear to what degree this approach could generalize to planning domains with more diverse plans, as it seems that the training and test problems are very similar.

Questions:

(Clarified in rebuttal) As acknowledged by the authors, only a single plan skeleton is considered here. The assumption seems to be that the sequencing (task-planning) stage has already identified the correct skeleton. However, the authors state that the learned models are "guiding" the task-level search. Am I misunderstanding this or is it misleading?

(Clarified in rebuttal + to be added to paper) If you were to consider multiple plan skeletons, how would the batch/forgetting algorithms be modified so that they do not spend the entire time sampling the wrong plan skeleton? If there are good ideas about this, then a passing mention in the paper would be appreciated.

(Clarified in rebuttal + to be added to paper) How much variance is there in the training data, since it's also estimated using sampling, i would imagine there is a certain amount of variance. Does it matter what sampling budget is used for backtracking during data collection?

Do you have any insight as to why the PF Attn models are doing so much worse than PF RNN?


**Summary Of Recommendation:**

I think that the idea being explored here is interesting, and worthwhile. I think the results are promising, but analysis/exposition of them is incomplete. I would love to see more experiments, and for the authors to provide more insight about what exactly we gain by using learning for culprit detection in this way.

(Updated after rebuttal) The additional experiments conducted after the initial review have led to what I believe to be a more thorough evaluation of the proposed methods and how they compare to reasonable baselines.

---

> ### Author Response · Authors · 2022-08-23
> **Response to Reviewer q4VZ (Part 1/3)**
>
> Please note that our response has been split into three parts due to the space constraint.
>
> Thank you for your valuable comments and time to review our paper. Your constructive comments will be very useful for improving our paper’s quality. There is one evaluation (i.e. effect of the sampling budget) we are conducting now to address the issue you raised.  But as it will take some time, we are sending you the main responses first, and the evaluation result later as soon as it is done. Please see our responses below that correspond to individual comments.
>
> ### Empirical evaluation with more baselines
> To the best of our knowledge, all sequence-before-specify-based TAMP planners in the literature rely on backtracking, and improving how backtracking is used in a novel way is the major value of the paper. Thus, we respectfully disagree that the insight into the fact that our method greatly outperforms backtracking is insufficient. Nonetheless, we do think the reviewer suggested baselines are interesting; we conducted experiments on the fixed-step backjumping methods and the method that always backjumps to a root node.
>
> We report the number of nodes visited in the search tree and the wall time, measured in seconds, obtained by the forgetting algorithm for solving each packing problem with 10 objects. We chose the packing task because the performance gap between backtracking and ours in packing is much smaller than that in NAMO, and thus the packing task generates a harder comparison. The planning code is executed on one core of Intel(R) Xeon(R) Gold 6342 CPU. Numbers represent the mean ± 95% confidence interval computed by solving 100 problems.
>
> |\# of fixed backjumping steps|\# of nodes visited|wall time|
> | ----------- | ----------- | ----------- |
> |1 (i.e. backtracking)|4414 ± 879|260.6 ± 58.1|
> |2|3093 ± 509|120.0 ± 21.1|
> |3|3323 ± 637|140.0 ± 26.9|
> |4|2892 ± 557|154.0 ± 26.6|
> |5|5408 ± 1106|391.6 ± 76.1|
> |6|8360 ± 1570|577.9 ± 109.6|
> |always backjumps to root|11212 ± 2002|685 ± 120|
>
> The results show that when the fixed step was set to 4, the baseline algorithm performed the best among all baselines. It also performed better than PF Attn but worse than other proposed models. Since PF Attn performed the worst among all proposed models, the above result implies that PF Attn needs further improvement in the model or the training process, or we treat PF Attn as another baseline. We will add the comparison results in the paper.
>
> ### Analysis on the overhead of the model inference
> Thank you for suggesting this. We agree this evaluation would be valuable. As this comment is closely related to the last comment in the issues (comment about the overhead of querying), we report them together.
>
> We report the ratio of model inference time over the total wall time (in %) for solving a single problem. The planning code is executed on the same machine in the above response and the IL/PF model is executed on one Nvidia A100 GPU. Numbers represent the mean ± 95% confidence interval computed by solving 100 problems.
>
> ||IL RNN|IL Attn|PF RNN|PF Attn|
> | ----------- | ----------- | ----------- | ----------- | ----------- |
> |Packing (%)|0.3 ± 0.2|2.1 ± 0.6|7.7 ± 2.7|1.2 ± 0.3|
> |NAMO (%)|12.5 ± 4.1|3.9 ± 1.4|12.4 ± 3.8|2.1 ± 0.6|
>
> The results show the overhead of the querying backjumping model is relatively cheap, consisting of less than 8% for the packing task. Even though querying takes 12.5% of the wall time for NAMO tasks, it is worthwhile considering that using backjumping reduces the total wall time from 6 ✕ 104 s to around 60 s. The results also show that the difference in the overhead of IL and that of PF is marginal.
>
> We also report the average time to determine the dead end when querying the model, measured in milliseconds. Numbers represent the mean ± 95% confidence interval computed by solving 100 problems
>
> ||IL RNN|IL Attn|PF RNN|PF Attn|
> | ----------- | ----------- | ----------- | ----------- | ----------- |
> |Packing (millisecs)|267.9 ± 217.5|56.5 ± 40.2|163.1 ± 112.4|16.9 ± 6.5|
> |NAMO (millisecs)|823.4 ± 462.2|292.7 ± 176.3|800.8 ± 433.9|138.1 ± 68.3|
>
> Even though the attention method is queried for every previous step while RNN is only queried once, the attention method is still much faster than RNN, because its time-series computation can be parallelized on GPUs while RNN needs to finish the time-series computation sequentially.
>
> We will add the results and the discussion into the paper as well.

---

> > ### Comment · Reviewer_q4VZ · 2022-08-27
> > **Empirical evaluation with more baselines**
> >
> > Thank you for taking the time to include these additional baselines. The results seem to suggest that the approach of backtracking a fixed number of steps (with that number tuned as a hyper parameter) is quite competitive (i.e. the 95% intervals overlap significantly) with the proposed learning method on the Packing task. If the authors believe that the NAMO task would present a more favourable comparison then I would recommend that you please add those results as well. As it is, I think it is a bit hard to justify the claim of "greatly" outperforming baselines. Alternatively, if the authors have some insight as to why one would prefer to use the proposed method rather than this much simpler baseline, please include this discussion. Thank you!

---

> ### Author Response · Authors · 2022-08-23
> **Response to Reviewer q4VZ (Part 2/3)**
>
> ### Generalization to diverse plans
> Please note that our approach is to learn a domain-specific heuristic from a domain-independent learning process, a technique in AI planning that aims to learn more effective heuristics than domain-independent heuristics. This means that our framework can be applied to any TAMP task, but the learning process learns a heuristic specific to a task that it currently works on. That is why we have similar training and task problems. The generalization capability comes from the learning model, and we acknowledge that our model is relatively primitive as it can only handle a varying number of objects with simple shape differences. Being able to deal with more realistic objects and different fixed objects (e.g. different shelves) is of our interest, as stated in the limitation section. However, even with these updates in the model, the training and test problems would still look similar (e.g. the model trained from the packing task won’t work for the NAMO task). Covering as many tasks as possible from a single trained model is an essential future direction for learning-driven TAMP, which would require a fundamentally different way of representing a state.
>
> ### Unclear on guiding at the task level
> Our TAMP planner does not necessarily output a single plan skeleton, and thus, there is no assumption that a plan skeleton found is always correct. Since it’s related to the next comment, we address it more thoroughly in the next response.
>
> Even if only a single plan skeleton is available, our learning method guides not at the motion level but the task level. When this plan skeleton has a length of K, then this consists of a sequence of K operations (or K actions). In the TAMP literature, finding a path for an individual operation requires motion-level reasoning, and finding a causal relationship among operations requires task-level reasoning. Our learning method guides the search, not within single motion planning (e.g. guiding samples to more promising regions in configuration space) but to decide which operation to handle by backjumping (e.g. the search skips Operations 3 and 4 and attempts Operation 2 to solve its motion planning problem). Thus, we say that our method guides at the task level. We hope this clarifies the confusion.
>
> ### Multiple plan skeletons
> We realize now that we missed explicitly addressing this critical point, so we thank the reviewer for pointing out this. As mentioned in the above response, our TAMP planner handles multiple plan skeletons as follows. In the sequencing stage, one plan skeleton is found (usually done by applying AI planning heuristics, e.g. Fast Downward). Then, in the specifying stage, the planner attempts to find a path by solving a sequence of motion planning problems. If the planner can’t find a path (i.e. the search tree is exhausted by backtracking or backjumping without finding a feasible solution), then the planner finds another plan skeleton in the sequencing stage (i.e. finding the second-best plan skeleton by AI planning heuristics). This process repeats until either the planner finds a solution, or all possible plan skeletons are exhausted (i.e. no solution).
>
> As the reviewer noticed, our explanation and evaluations are done for a single plan skeleton (this is mainly for the sake of ease of presentation). Extending to multiple plan skeletons is straightforward; we learn a plan skeleton-specific model. Our algorithms only need a minor change, applying a model that matches with a corresponding plan skeleton. Note that there will be only a finite number of plan skeletons, and similar plan skeletons will generally be obtained as problem data is  drawn from a fixed distribution. Designing a model agnostic to plan skeletons may be possible, but we expect that’s an extremely difficult learning problem. We included this explanation in the paper.

---

> > ### Comment · Reviewer_q4VZ · 2022-08-27
> > **Reviwer respone**
> >
> > Unclear on guiding at the task level:
> >
> > Thank you for clarifying. Is it fair to say that the proposed approach provides guidance to the "specifying" procedure while considering a particular plan skeleton, but does not provide any guidance to the high level planner as to which plan skeleton to consider next (in the event of failure)?
> >
> > Multiple plan skeletons:
> >
> > Thank you for clarifying how the approach could work when multiple plan skeletons need to be considered. Could you clarify why it is that the number of plan skeletons is finite? Are we assuming that each object can only be interacted with once?

---

> ### Author Response · Authors · 2022-08-23
> **Response to Reviewer q4VZ (Part 3/3)**
>
> ### Variances in the training data and effect of the sampling budget
> Thank you for an insightful question.
>
> (1) Variances in the training data:
>
> In our planner, sampling approximates a continuous domain in a discrete way. Thus, using different numbers of samples generates sequences of different discrete spaces. For example, when N=10 and 100, 10 and 100 samples are found, respectively, from a uniform distribution, and the planner repeats that for the plan length of K. Because of this, direct comparison between partial plans at a dead-end when using different numbers of samples is uninformative. Instead, we analyze “false-negative ratios” for different numbers of samples when the task is to find a feasible placement in the packing as a proxy for variance. False-negative ratios measure the contribution to noisy labels and indicate a sufficient number of samples not to miss a feasible placement for a given problem domain. Note that we can obtain noiseless data if an infinite number of samples is used.
>
> |sample size|10|30|50|70|90|
> | ----------- | ----------- | ----------- | ----------- | ----------- | ----------- |
> |false-negative ratio|0.72|0.50|0.34|0.21|0.10|
>
> We report the results of finding a feasible placement for the 10-th object in the packing problem when there are 9 objects in the cabinet. The results imply that if we use more than 90 samples per level, we are unlikely to miss a feasible case at each level of the search tree. However, note that there exist infeasible cases during the search (e.g. any placements are infeasible due to other objects blocking the cabinet entrance). Since infeasibility is unknown in advance in motion planning, the search must spend all 90 samples for infeasible cases. Therefore, increasing the number of samples has a trade-off between finding a solution from feasible cases and wasting a lot of time on infeasible cases. The effect of the trade-off would vary depending on the problem.
>
> (2) Effect of the sampling budget:
>
> This is a good point and worth evaluating. We are working on training the models with different numbers of samples and evaluating the packing task. We will report the results soon.
>
> ### Why does PF Attn perform much worse than PF RNN?
> This is because PF RNN is more sample-efficient than PF Attn. As described in Sec. 4.2 and 6.2, when deciding the culprit, PF needs to query the model for each of K previous steps and decides based on K outputs. Hence, the PF method is more sensitive to model accuracy. In relatively low-data evaluations, like Packing with 500 problems and NAMO with 250 problems, PF RNN is more sample-efficient and has better accuracy than PF Attn, thus achieving better culprit prediction accuracy and shorter planning time.
>
> ### Not well-described evaluation tasks
> Thank you for the clarification question. The reviewer is correct that initial poses of movable objects are sampled at random. We added this explanation in the evaluation tasks section.

---

> ### Author Response · Authors · 2022-08-26
> **Additional evaluation for the effect of the sampling budget**
>
> We conducted additional evaluations on the performance of learning models in the packing task when varying the number of samples (i.e. N in the paper) used in training. Specifically, we set the number of samples to be 10, 30, and 50 in training while that to be 30 in testing.
>
> |# of samples in training / # of samples in testing|IL RNN|IL Attn|PF RNN|PF Attn|
> | ----------- | ----------- | ----------- | ----------- | ----------- |
> |10 / 30|1889.6 ± 328.0|2153.1 ± 369.8|3276.7 ± 597.4|2932.5 ± 450.3|
> |30 / 30|2464 ± 464|2638 ± 602|2367 ± 517|2913 ± 569|
> |50 / 30|5752.6 ± 1446.3|6672.3 ± 1819.5|4859.6 ± 1100.5|4267.2 ± 1035.8|
>
> The results show that for IL 10/30 performed the best and for PF 30/30 performed the best, while for 50/30 both performed the worst. It is expected that at the extreme (N=1) in training, IL would predict to backjump to near the root as most of the time, a feasible placement wasn’t found with N=1. This would behave similarly to a baseline, always backjumping to a root; thus, its performance is expected to be poor. In the “variances in the training data” response, we stated that comparing different numbers of samples in testing is uninformative. Therefore, we conclude that for a given number of samples in testing, we can treat the sample size in training as a hyperparameter, and one can tune it for their applications for better performance. We included this explanation in the supplementary document.

---

> ### Author Response · Authors · 2022-08-27
> **Response to follow-up comments**
>
> Thank you for your additional comments and follow-up questions.
>
> ### Empirical evaluation with more baselines
>
>  We do agree with the reviewer that one of the baselines (i.e. 4-step backjumping) performs competitively with the proposed models; finding the best-fixed step from training data can be considered another learning-based baseline, which is simpler than our models. However, we want to point out that we obtained the initial results in Table 1 just enough to show their surpassing performance over backtracking, not over new baselines. Given limited rebuttal time, we couldn't redo the entire evaluations for Table 1 (which we plan to do). But we have one evidence of our models outperforming new baselines; when varying the number of samples in training (in the "Additional evaluation for the effect of the sampling budget" response), we observe that IL greatly outperforms new baselines (we didn't observe the same for PF, though). This evidence and different training settings (such as more training problems and more episodes) imply there still is room for the models to improve their performances further. We will include more optimized results of our models in the paper.
>
>  As the reviewer suggested, we will also include the new comparison results for NAMO and the discussion in the paper. Thank you again for this insightful comment!
>
>  ### Unclear on guiding at the task level
>
>  That is precisely correct. Reasoning over different plan skeletons (i.e. more high-level reasoning) is an interesting direction, which we haven't seen studied in the literature to the best of our knowledge.
>
>  ### Multiple plan skeletons
>
>  We do not assume that each object can be interacted with only once. An object is allowed to be interacted with as many times as the robot wants. But please notice that any solutions obtained by solving a planning problem would contain a finite number of actions, thus, a finite number of interactions with objects. If a planning problem has a solution containing infinite actions, then we think the problem is unsolvable. Nonetheless, the reviewer's point is interesting. Since any problems have finite numbers of interactions, operations, and different objects, the number of all possible combinations of plan skeletons is finite.

---

### Official Review · Reviewer_MGEc · 2022-07-31

**Originality:** Fair
**Technical Quality:** Fair
**Clarity Of Presentation:** Good
**Impact:** 3

**Recommendation:**

Weak Accept: I recommend accepting the paper, but will not argue for my recommendation if the majority of other reviewers have a different opinion.

**Summary:**

This paper focuses on improving a task and motion planning approach. Due to the infeasibility of covering the search space in such problems extensively, the solver might get stuck in later stages of the search due to some earlier assignment of the world/object configurations. This work proposes to train a prediction model to backjump onto the search state where the bottleneck might have occurred. In essence, the default backtracking is replaced with this backjumping heuristic. The learned model is shown to improve the computational efficiency of the solver on two different tasks.

**Issues:**

- as mentioned in the 'weaknesses' above, the most critical issue is the experimental setup: same container, same object types, simpler shapes, same arrangement of fixed objects, etc. and using the full-state knowledge -> they do not help the reader accept the effectiveness of the proposed addition of a backjumping predictor as useful for real sequential manipulation tasks and the presented results as convincing. Yes, the authors already acknowledge those, but I think these are still serious limitations that without taking some steps towards handling more realistic scenarios, the value of the work will be highly limited.
- varying training data size: why increasing the data size 4-fold doesn't improve the prediction efficiency?
- what are the computation times (wall-clock time)?
- related work: there is some line of work ([R1] + some follow-up) that investigates learning heuristics to speed up TAMP solvers. How do you compare to them?
- related work: As mentioned by the authors, joint optimization based approaches can also solve such issues [12] (there is also LGP-line of work which have been almost totally neglected in the paper). More discussion might help emphasize the difference to those, and the advantages of the approach used within this paper.

[R1] D. Driess et al., "Deep Visual Heuristics: Learning Feasibility of Mixed-Integer Programs for Manipulation Planning", IEEE ICRA, 2020.


**Quality Of The Limitations Section:**

Additional details required

**Reviewer Expertise:**

3: The reviewer is fairly confident that the evaluation is correct

**Robotics Focus:**

Relevant but unlikely to deploy to hardware in near future

**Strengths And Weaknesses:**

strengths:
- the proposed predictor is effective at solving similar problems being tested

weaknesses:
- full state knowledge is assumed, and
- same/similar setups are used for both tasks, e.g., simple and same object geometries: might then use user defined heuristics and/or analytical methods to compute feasibility?

**Summary Of Recommendation:**

The paper implements a learning architecture to predict where to backjump in the search process when the solver gets stuck during solving for a task and motion planning problem. As a result, instead of backtracking, which is the default procedure, the predictor allows instantly jumping to earlier levels of the search tree, which in turn increases the computational efficiency. Both of the scenarios presented are relatively simplistic due to the simplifying assumptions. Thus, the work falls short on justifying the efficiency and applicability of the proposed inclusion of a predictor for a broader set of sequential manipulation tasks, especially to the more realistic ones.

---

> ### Author Response · Authors · 2022-08-22
> **Response to Reviewer MGEc (Part 1/2)**
>
> Please note that our response has been split into two parts due to the space constraint.
>
> Thank you for your valuable comments and time to review our paper. Your constructive comments will be very useful for improving our paper’s quality. Please see our responses below that correspond to individual comments.
>
> ### Simple experimental setup
> We fully agree with the reviewer that the simplifying assumptions in the scenarios considered in this paper prevent direct application to more complex, real-world scenarios. If we understand correctly, the reviewer is questioning whether it’s possible to make an interesting and valuable contribution to TAMP without embracing the full suite of real-world complexities that are the focus of recent learning-based work.  We respectfully submit that the focus of this paper on a relatively austere scenario is necessary to isolate the planning-related issues that are the main subject of this paper. After establishing the potential utility of backjumping under these assumptions, as we do in this paper, it will be an important next step to extend the methods to more complex and uncertain domains, presumably with the need for significantly more data and computational resources. However there is not room to give that study adequate treatment in the same conference paper that introduces the fundamental backjumping approach.
>
> As justification for this paper representing a significant contribution on its own, we offer two different perspectives as follows.
>
> (1) Planning perspective:
> Our work is based on the state-of-the-art general-purpose TAMP planner, which can handle any high-dimensional hybrid search problems without modifications. In the AI planning community, even low-dimensional “discrete” search problems are known to be PSPACE-complete. This means that the search problems we consider in the paper are difficult, even without any uncertainties. Considering uncertainties makes the planning formulation extremely complicated (as it requires belief space planning), and it is still an active area of research; thus, there is no generally accepted formulation. That is why many recent TAMP papers with the same assumptions as ours are valued by the community: they address new and different challenges in TAMP (e.g. the G-TAMP paper). Moreover, since we use a sampling method for motion planning, analytical methods are generally unavailable. We expect that the same principle proposed in the paper may prove to be applicable to those harder problems. Our work is a proof-of-the-concept in a more limited formulation to focus on the backjumping idea.
>
> (2) New learning problem in TAMP:
> There are several attempts leveraging machine learning techniques in TAMP problems, such as learning to guide sampling and learning to predict feasibility (e.g. the paper [1] that the reviewer referred to). Compared with existing work focusing on learning at the “motion level,” the main value of our work is a novel way of leveraging learning in TAMP, “task-level” heuristic learning, which has not been studied in the literature. As this paper is about learning from hard problems within the SOTA TAMP planner, we think it’s very important to cover every detail precisely before attempting more realistic settings, which could make the paper more superficial.
>
> [1] D. Driess et al., "Deep Visual Heuristics: Learning Feasibility of Mixed-Integer Programs for Manipulation Planning", ICRA, 2020.
>
> ### Why increasing PF dataset from 500 problems to 2000 problems doesn’t improve its performance
>
> Thank you for pointing out this. We have several hypotheses on this result, which we added in the supplementary document.
>
> (1) Even from a small number of problems, PF can collect many labels, and thus further increasing data size doesn’t significantly affect the performance. Specifically, PF can collect one feasibility likelihood label for each node visited in the search tree (see details in Sec. 4). As a result, PF already has many training labels even with small data size, and it thus improves less than IL when trained with more data.
>
> (2) Note that the PF model is not trained to predict a backjumping node directly; instead, it makes up to K inferences (proportional to a partial plan length), and we pick one with the lowest PF likelihood outside the model as a backjumping node indirectly. Because of this, increasing data size does not necessarily lead to improved end result performance. Designing a model that learns to predict a backjumping node based on PF likelihoods predicted as an auxiliary task would be an interesting direction.
>
> (3) There is some stochasticity in both model selection (training checkpoint) and evaluation. We select the models based on a randomly sampled validation dataset. The selected model could happen to predict well on the validation data and thus doesn’t improve on the test data.

---

> > ### Author Response · Authors · 2022-08-22
> > **Response to Reviewer MGEc (Part 2/2)**
> >
> > ### Computation time (wall-clock time)
> >
> > The wall time for solving each problem, measured in seconds, is below. The wall time is roughly proportional to the number of nodes visited in the search tree.
> > The planning code is executed on one core of  Intel(R) Xeon(R) Gold 6342 CPU and the IL/PF model is executed on one Nvidia A100 GPU. Numbers represent the mean ± 95% confidence interval computed by solving 100 problems.
> >
> > |  | Backtracking | IL RNN | IL Attn | PF RNN | PF Attn |
> > | ----------- | ----------- | ----------- | ----------- | ----------- | ----------- |
> > | Packing | 202.0 ± 45.4 | 97.4 ± 19.3 | 162.1 ± 32.7 | 94.7 ± 22.0 | 122.0 ± 23.8 |
> > | NAMO | 61754.3 ± 24608.4 | 66.0 ± 19.9 | 67.8 ± 19.9 | 58.7 ± 18.3 | 11648.3 ± 12342.7 |
> > |  |  |  |  |  |  |
> >
> > We will add the results and the discussion into the paper as well.
> >
> > ### Related work ([1] that learns heuristics to speed up TAMP solvers and LGP line of work)
> > Since [1] is one of the LGP papers, we respond to two related questions together.
> >
> > (1) Connection to LGP:
> >
> > We indeed had a paragraph in the related work section devoted to the connection with optimization-based TAMP, including LGP, but decided to remove it to meet the maximum page length requirement. We prioritized it below other points because there is no clear comparative criterion to distinguish between optimization-based planners and sampling-based planners as they have fundamentally different formulations and characteristics. Like in motion planning, one says that optimization-based finds a smooth path but has a chance to get trapped in local minima, whereas sampling-based is good at dealing with a high-dimensional search space but can be slow if a narrow passage exists. However, how best to make a direct comparison between the two is unclear; one may work better than another depending on a task. For these reasons, a heuristic learning method proposed for one may not apply to another in the first place (e.g. our method relies on samples obtained from uniform sampling, and it is not an optimization but a feasibility problem). It is an interesting open question to design motion planning agnostic heuristics that work for both.
> >
> > (2) Other learning heuristic papers in TAMP (such as [1]):
> >
> > The reference [1] is about learning to predict feasibility at the motion level in TAMP (similar idea from sampling based is [2]). The reference [3] learns to guide sampling in TAMP. Other heuristic learning methods in motion planning may be applied to TAMP as well, such as learning to guide sampling [4] and again learning to predict feasibility [5].
> >
> > As pointed out in the above response, our approach learns task-level heuristics while others learn motion-level heuristics. Apart from this difference, we do not see different approaches competing but complementing each other (unless one tries to use different methods together from the same approach, e.g. [1], [2], and [5]). Ideally, we want our planner to predict feasibility so that it can terminate in infeasible problems, guide sampling to find promising regions, backjump to a culprit variable when meeting a dead end, and so on, improving the overall planning efficiency. An analysis on this combination of approaches is an important future research.
> >
> > We added the above explanation in the related work section.
> >
> > [2] Wells, A.M., et al, Learning feasibility for task and motion planning in tabletop environments. RA-L, 2019.
> >
> > [3] Chitnis, R., et al, Guided search for task and motion plans using learned heuristics. ICRA, 2016.
> >
> > [4] Zhang, et al, Learning implicit sampling distributions for motion planning. IROS, 2018.
> >
> > [5] Li, S. and Dantam, N.T., Learning proofs of motion planning infeasibility. RSS, 2021.

---

### Official Review · Reviewer_Kqwz · 2022-07-31

**Originality:** Very Good
**Technical Quality:** Very Good
**Clarity Of Presentation:** Good
**Impact:** 3

**Recommendation:**

Strong Accept: I recommend accepting the paper and will argue for my recommendation even if other reviewers hold a different opinion.

**Summary:**

The authors propose to adapt the backjumping technique used in constraint solving. Since locating the culprit action is not analytically tractable, they instead propose learning a domain specific model to predict the correct location to jump to. They find that this leads to significant reductions in search time on all tasks tested, including a 99% reduction in search time on the NAMO task.

**Issues:**

~~Question of what happens when the model is inaccurate is inadequately addressed~~

**Quality Of The Limitations Section:**

Additional details required

**Reviewer Expertise:**

2: The reviewer is willing to defend the evaluation, but it is quite likely that the reviewer did not understand central parts of the paper

**Robotics Focus:**

Highly relevant to robotics but no hardware experiments

**Strengths And Weaknesses:**

This is an interesting way of improving efficiency with impressive results. It offers a clever way of encoding the problem as well, which allows for generalization across an interesting domain of problems.

One question I felt was inadequately addressed was what happens when the model makes an incorrect prediction. I presume that if the model under-predicts the length of the optimal jump, then it will continue to perform backtracking. However, it's unclear what happens if the model over-jumps. Does it prevent an optimal solution? Can it prevent the solver from finding any solution? How often does the model over-predict a jump? I see that some of these questions are addressed in the appendix, but I think it deserves at least some comment in the main body of the paper.

**Summary Of Recommendation:**

~~The paper could be clearer in what the tradeoffs of using backjumping guided by a learned model are and how accurate the model is. However, these are somewhat minor issues in my opinion, and it is a solid idea backed by impressive experimental results, so I recommend to accept.~~

The authors addressed my central concerns in their response.

---

> ### Author Response · Authors · 2022-08-22
> **Response to Reviewer Kqwz**
>
> Thank you for your valuable comments and time to review our paper. We are encouraged that the reviewer thinks highly of our paper. Please see our response below corresponding to the question you asked.
>
> ### When the model makes an incorrect prediction
> Thank you for your question. That is indeed a critical point, and that’s why we analyzed and included it in the supplementary document. However, unfortunately, the analysis of prediction accuracy is not as simple as it looks because the process involves sampling that approximates a continuous domain in some arbitrary discrete way (i.e. uniform sampling). When the prediction under-jumps (GT in the paper), sampling won’t be able to find a feasible solution without additional backtracking or backjumping as the culprit variable exists at a higher node in the tree; the reviewer’s understanding is correct. Let’s now think of when the prediction over-jumps (LT in the paper). In this case, the search misses the culprit variable as the reviewer pointed out. Still, because we use sampling, the search may find a new opportunity to find a feasible solution by sampling a different value for the variable that used to be the culprit before backjumping but is no longer a culprit. As our work does not focus on learning a sampling strategy, it is also possible that the above situation may make a new culprit variable by sampling a wrong value. Table 3 in the supplementary document shows how frequently both types of predictions occur.  Mitigating the potential lack of completeness due to the second type of prediction is an interesting open direction.
>
> Due to this complex stochastic phenomenon, we think it is hard to make an insightful conclusive statement about the effect of two predictions except that backjumping as close as possible to a culprit variable is desirable. To meet the maximum page length requirement, the full treatment of this issue will only fit in the supplementary material.  But we will attempt to summarize it concisely in the main document as well.

---

### Official Review · Reviewer_MW4W · 2022-08-01

**Originality:** Good
**Technical Quality:** Very Good
**Clarity Of Presentation:** Very Good
**Impact:** 4

**Recommendation:**

Strong Accept: I recommend accepting the paper and will argue for my recommendation even if other reviewers hold a different opinion.

**Summary:**

This paper proposes a data-driven approach to efficient task-level search in geometric task and motion planning (G-TAMP). Specifically, the paper replaces inefficient backtracking with backjumping, which directly backtracks multiple steps in the search tree when an infeasible level is detected. The authors present two supervised learning methods to learn to detect the “culprit action” (i.e. the earliest action in the search tree that needs to be corrected by backjumping in order to resolve infeasibility). The first approach is imitation learning, where the tree level corresponding to the culprit action is directly output based on the state sequence. The second approach is classification, in which the feasibility of a partially-refined future plan is checked from a fixed state at some level of the tree. The effectiveness of both approaches are demonstrated in two different mobile manipulation tasks in a physics-based motion simulator.

**Issues:**

~~Please refer to the weaknesses section and the questions stated above.~~

**Quality Of The Limitations Section:**

Limitations are addressed clearly

**Reviewer Expertise:**

3: The reviewer is fairly confident that the evaluation is correct

**Robotics Focus:**

Highly relevant to robotics but no hardware experiments

**Strengths And Weaknesses:**

**Strengths of the Paper**

1) Tree search in continuous state spaces is notoriously difficult to solve. Yet, it appears quite often in robotics, including the G-TAMP problems that are addressed in this work. The paper proposes an interesting approach to accelerating tree search algorithms by means of learning a backjumping strategy. The general idea of using machine learning for tree search is not new (see the weaknesses section below), but the proposition of backjump learning is novel as far as I am aware.
2) Even though no theoretical analysis is provided, the empirical evaluation of the proposed approaches are well-executed and convincing.
3) The quality of presentation is high in general, although some improvements could be made (see the weaknesses section below).


**Weaknesses of the Paper**
1) (*Resolved after rebuttal*) As noted above, the general idea of data-driven tree search for improved efficiency is not new. For instance, Masti and Bemporad [1] propose to use a learned predictor to circumvent costly branch-and-bound pruning in exploring a binary search tree, in the context of mixed integer programming. In [2], Pinto and Fern present a supervised learning approach to efficient guided-search in a Markov-decision tree. These approaches can be loosely categorized as “learning an efficient sampling (or branching) strategy in tree search,” which would be an alternative domain-independent way of learning domain-specific heuristics. In contrast, the method presented in this paper fixes a sampling strategy (i.e. either batch of forgetting sampling) and instead learns how to correct the plan from bad samples. This observation leads to the following fundamental, motivational question: “how is the backjumping correction strategy advantageous over directly learning an efficient sampling strategy (which seems more common in data-driven, accelerated tree search) in the first place?” This point should be clarified in the paper.

2) (*Resolved after rebuttal*) In terms of the presentation quality, I was a bit confused by Section 2.2 which lacks the details of how the sampling is performed, until I saw the two sampling methods described finally in Section 5. The authors may want to re-order or augment the explanations about sampling for extra clarity.

[1] Masti, Daniele, and Alberto Bemporad. "Learning binary warm starts for multiparametric mixed-integer quadratic programming." In 2019 18th European Control Conference (ECC), pp. 1494-1499. IEEE, 2019.

[2] Pinto, Jervis, and Alan Fern. "Learning partial policies to speedup MDP tree search via reduction to IID learning." The Journal of Machine Learning Research 18, no. 1 (2017): 2179-2213.

**Other Minor Points and Questions**

1) (*Resolved after rebuttal*) How is the plan length K selected? Is that chosen heuristically by domain knowledge? It seems that the choice of K is crucial in long-horizon planning tasks, since underestimating the value of it could lead to the robot failing to find any feasible plan regardless of the backjumping strategy.
2) (*Resolved after rebuttal*) In Table 4, the imitation learning approach seems to benefit more from increased data size than the classification approach. Is there a reasonable explanation for that?
3) (*Resolved after rebuttal*) In tables, how specifically is “statistically significantly different from one with *” determined?
4) (*Resolved after rebuttal*) Typo: “On important practical challenge” → “One important practical challenge” (Section 1, line 17)
5) (*Resolved after rebuttal*) Typo: “Often done often” → “Often done” (Section 1, line 26).



**Summary Of Recommendation:**

~~I recommend "weak accept," based on the overall novelty of backjumping learning (as noted in the strengths section) as well as the motivational question of "why learn backjumping instead of learning a sampling strategy itself?" (as noted in the weaknesses section).~~

My concerns have been addressed after the rebuttal. I am convinced that the overall novelty of the paper is high, providing an interesting alternative to guided sampling for accelerated search. Thus I recommend acceptance.

---

> ### Author Response · Authors · 2022-08-22
> **Response to Reviewer MW4W**
>
> Thank you for your valuable comments and time to review our paper. Your constructive comments will be very useful for improving our paper’s quality. Please see our responses below that correspond to individual comments.
>
> ### Existing research on learning an efficient sampling strategy in tree search
>
> Thank you for suggesting an interesting alternative direction comparable to our approach.  We agree that learning an improved sampling strategy from past experience is promising, and we are aware that there is existing work in motion planning that addresses this direction, such as [1], although it is a single-mode motion planning problem.  In fact the idea of learning search control knowledge is at least as old as the PRODIGY system [1] from more than 30 years ago.  The referred papers (one by Masti and Bemporad and another by Pinto and Fern) consider MIQP problems and MCTS, which are optimization problems, whereas our problem considers hybrid CSP, which is a feasibility problem. Since there is a fundamental difference between those problems, it is unclear how to take advantage of the methods of the referred papers for our problem, although it may be doable. On the other hand, we do think that the improved sampling strategy proposed by [2] can directly be applied to individual actions in a plan skeleton, improving the entire tree search efficiency.  This is a promising direction for future work and will be identified as such in the paper.
>
> However, we would like to point out that learning a sampling strategy and learning backjumping heuristics do not compete with each other, but they can coexist to improve planning efficiency further; They complement each other by guiding sampling to more promising regions while backjumping to a culprit action when meeting a dead end. The paper’s value that we see is the novel way of leveraging machine learning in TAMP. Besides these two, there are also different approaches, such as learning to predict feasibility, learning hyperparameters of a planner, etc., all of which can coexist with, and perhaps even synergize with, our approach. We see the ultimate goal for learning-leveraged TAMP as being unifying all of these different approaches into a single system, and thus, it is an interesting future direction. We added the above explanation in the related work section.
>
> [1] Veloso, M., et al, Integrating planning and learning: The PRODIGY architecture. Journal of Experimental & Theoretical Artificial Intelligence, 1995.
>
> [2] Zhang, C., et al, Learning implicit sampling distributions for motion planning. IROS, 2018.
>
> ### How sampling performs is confusing in Section 2.2
>
> Thank you for pointing out this. Finding values of c_k is generally done by uniform sampling from its domain in the motion planning literature to prove the probabilistic completeness (e.g. [3]). We added this in Section 2.2.
>
> [3] Hauser, K. and Latombe, J.C., Multi-modal motion planning in non-expansive spaces. IJRR, 2010.
>
> ### Plan length K
>
> Please note that the tuple in the G-TAMP definition does not include K, which means K is not an input. Rather, K is an output from the planner; the TAMP planner finds a solution without having to know a plan length a priori and the solution's plan length is defined as K. We introduce K as a variable to explicitly define the number of choices for backjumping.
>
> ### IL benefited more from increased data size than PF
>
> This is because, even from a small number of problems, PF can collect many more labels than IL, and thus further increasing data size has smaller effects on PF than on IL. Specifically, while IL only collects one label for each dead-end, PF can collect one feasibility likelihood label for each node visited in the search tree (see details in Sec. 4). As a result, PF already has many training labels even with a small data size, and it thus improves less than IL when trained with more data.
>
> However, even though PF has many more labels than IL from the same number of problems, it still performs worse than IL because of its iterative way of determining the culprit. Specifically, when determining the culprit from all previous K steps before the dead-end, IL only needs to query the model once, but PF needs to query the model for each previous step and decides based on K outputs (see details in Sec. 4 and Sec 6.2). In other words, if any of K outputs is wrong, PF could estimate the culprit wrongly. As a result, even though PF has the data advantage, it still performs worse than IL.
>
> With more training data, though, the accuracy of a single PF model query is improved; the culprit estimation may improve less because of this K-step aggregation. We added this explanation in the appendix.
>
> ### Statistically significantly different from one with *
>
> We say two values are not statistically significantly different from each other if their 95% confidence intervals are overlapped. We added this explanation for clarity.
>
> ### Typos
>
> We fixed them. Thank you!

---

> > ### Comment · Reviewer_MW4W · 2022-08-26
> > **Response to Authors**
> >
> > Thank you for carefully addressing my comments. Your response answers my question of why learn backjumping instead of a sampling strategy, and I agree that backjumping and guided sampling can coexist. I will increase the impact from 2 to 4 in Part 2 of the review process.

---

### Meta-Review · Area_Chair_bqNT · 2022-08-14

**Recommendation:** Accept (Poster)
**Confidence:** 5

**Metareview:**

The paper proposes to learn backjumping heuristics for geometric task and motion planning frameworks. The key idea is to identify the "culprit action" that leads to the infeasible path during the search and directly "backjump" to correct the action selection. The authors present two types of solution under this idea: (1) directly predict the culprit action and (2) estimate the plan feasibility given each action choice. The method is tested on two types of rearrangement tasks using a mobile platform in simulation.

Strengths: The reviewers agree on the importance of improving the efficiency of the search algorithm for large-scale manipulation planning problems, and that learning is a promising path to this objective. They also agree on the empirical effective of the proposed method, at least based on the evaluation setup presented in the paper.

Weaknesses: The main comments focus on empirical evaluations: (1) lack of comparison to baselines (R-q4VZ) such as guided sampling (R-MW4W) and (2) lack of more realistic and diverse evaluation tasks (R-q4VZ, R-MGEc) and ablation studies (R-Kqwz, R-q4VZ)

Post-response period: The authors have addressed most of the concerns raised by the reviewers. In particular, they have included additional baseline results, runtime performance, and answered most conceptual questions. I also agree that the method provides a conceptually simple yet effective improvement to a core component in TAMP.  However, during the post-response period, some doubts still remain on the applicability of the method on a broader set of tasks. Based on the response and the updated ratings (as well as their justifications), the AC recommends that the paper should be accepted with a poster presentation.

**Best Paper Nomination:**

No

---

> ### Author Response · Authors · 2022-08-26
> **Response to Area Chair bqNT**
>
> We thank Area Chair and Reviewers for their valuable comments and time to review our paper. We have carefully addressed the reviewers’ comments, which improved our paper’s quality significantly. Regarding the comments on the weaknesses, please refer to our individual responses to the reviewers as follows.
>
> - **Lack of comparison to baselines**: Please see “Empirical evaluation with more baselines” from Response to Reviewer q4VZ (Part 1/3).
> - **Guiding sampling**: Please see  “Existing research on learning an efficient sampling strategy in tree search” from Response to Reviewer MW4W.
> - **Lack of more realistic and diverse evaluation tasks**: Please see  “Simple experimental setup” from Response to Reviewer MGEc (Part 1/2), “Empirical evaluation with more baselines” from Response to Reviewer q4VZ (Part 1/3), and "Generalization to diverse plans" from Response to Reviewer q4VZ (Part 2/3).
> - **Lack of ablation studies**: Please see  “When the model makes an incorrect prediction” from Response to Reviewer Kqwz, and “Empirical evaluation with more baselines” from Response to Reviewer q4VZ (Part 1/3).